# New Insights into Avian Infectious Bronchitis Virus in Colombia from Whole-Genome Analysis

**DOI:** 10.3390/v14112562

**Published:** 2022-11-19

**Authors:** Gloria Ramirez-Nieto, Daiana Mir, Diego Almansa-Villa, Geovanna Cordoba-Argotti, Magda Beltran-Leon, Nelida Rodriguez-Osorio, Jone Garai, Jovanny Zabaleta, Arlen P. Gomez

**Affiliations:** 1Grupo de Investigación en Microbiología y Epidemiología, Facultad de Medicina Veterinaria y de Zootecnia, Universidad Nacional de Colombia, Bogotá 111321, Colombia; 2Unidad de Genómica y Bioinformática, Departamento de Ciencias Biológicas, Centro Universitario Regional Litoral Norte, Universidad de la República (UdelaR), Salto 50000, Uruguay; 3Stanley S. Scott Cancer Center, School of Medicine, Louisiana State University Health Science Center, New Orleans, LA 70112, USA; 4Department of Interdisciplinary Oncology, School of Medicine, Louisiana State University Health Science Center, New Orleans, LA 70112, USA

**Keywords:** *Gammacoronavirus*, genome sequencing, poultry, respiratory tract diseases

## Abstract

Infectious Bronchitis (IB) is a respiratory disease caused by a highly variable *Gammacoronavirus*, which generates a negative impact on poultry health worldwide. GI-11 and GI-16 lineages have been identified in South America based on Infectious Bronchitis virus (IBV) partial S1 sequences. However, full genome sequence information is limited. In this study we report, for the first time, the whole-genome sequence of IBV from Colombia. Seven IBV isolates obtained during 2012 and 2013 from farms with respiratory disease compatible with IB were selected and the complete genome sequence was obtained by NGS. According to S1 sequence phylogenetic analysis, six isolates belong to lineage GI-1 and one to lineage GVI-1. When whole genome was analyzed, five isolates were related to the vaccine strain Ma5 2016 and two showed mosaic genomes. Results from complete S1 sequence analysis provides further support for the hypothesis that GVI-1, considered a geographically confined lineage in Asia, could have originated in Colombia. Complete genome information reported in this research allow a deeper understanding of the phylogenetic evolution of variants and the recombination events between strains that are circulating worldwide, contributing to the knowledge of coronavirus in Latin America and the world.

## 1. Introduction

Avian Infectious Bronchitis (IB) is an acute and highly contagious respiratory viral disease that affects poultry of all ages worldwide. IB is caused by the Infectious Bronchitis Virus (IBV), which affects mainly the upper respiratory tract and can also spread to non-respiratory tissues causing lesions in the renal and reproductive systems [1]. IB disease affects both broiler and layer chickens, causing enormous economic losses in poultry production systems. Due to its socioeconomic and avian health impact related to mortality and decreased egg production, IB is on the World Organisation for Animal Health (WOAH) list of notifiable diseases.

IBV is a member of the order Nidovirales, family Coronaviridae, subfamily Orthocoronavirinae, genus *Gammacoronavirus*, which primarily infects birds along with *Deltacoronaviruses*. It is an enveloped virus with a diameter of about 120 nm and a pleomorphic shape with spikes of about 20 nm. Its single-stranded positive-sense RNA genome is about 27.5–28 kb long and encodes four structural proteins (S, M, N, and E), two polyproteins (1a and 1b), and four accessory proteins (3a, 3b, 5a, and 5b), arranged from the 5’ to the 3’ end as follows: 1a-1b-S-3a-3b-E-M-5a-5b-N [2,3].

ORFs 1a and 1b represent about 60% of the genome. These are cleaved into 15 non-structural proteins (nsp2-nsp16) implicated in the viral replication process through RNA-dependent RNA polymerase. The protein E (Envelope) is an integral membrane protein that, along with the protein M (Membrane), interacts for viral assembly with the protein N (Nucleocapsid), which is also responsible for viral genome encapsidation. The polyproteins 3a, 3b, 5a, and 5b are non-structural proteins whose functions are not well-known [4].

Among the structural proteins, the Spike (S) protein plays a key role in receptor recognition and is divided into two functionally distinct subunits, S1 and S2. The S2 subunit is conserved between IBV strains and participates in membrane fusion. Meanwhile, S1 is involved in antigen neutralization, hemagglutination, and cell tropism determination [1,2]. Three hypervariable regions (HVR) located within amino acids 38–67, 91–141, and 274–387 are present in the S1 subunit. The S1 high mutation and recombination rate leads to a variability associated with the frequent appearance of new genotypes and antigenic variants worldwide, resulting in low or no cross protection in infected birds. Hence, mutations within this genome region may result in the emergence of new viral variants. However, an analysis based on the complete genome sequence is necessary to fully understand evolutionary and recombination events in IBV.

Currently, six IBV genotypes (GI-GVI) have been classified through phylogenetic inference, based on the S1 complete nucleotide sequence, with the GI genotype being the most diverse, and further sub-classified into 27 lineages [5]. From these, one main genotype G1 and two lineages (11 and 16) have been reported in South America [6,7,8,9,10]. Even though S1 sequence analysis is the most widely used method for IBV strain characterization, there is also a serotype-based classification system, according to the antigenic features of the isolates, resulting in different serotypes such as Massachusetts (Mass), Connecticut (Conn), 793/B, Italy-02, and Qx, among others [11,12,13].

In Colombia, the first isolation of IBV was reported in 1963 from samples of 25-day-old broilers and five-month-old layers with respiratory signs [14]. The control of IB is based on vaccination schemes with live attenuated and/or inactivated vaccines related to the Mass serotype. Even though they are not currently in use, a few vaccines, based on D274/D207, Ark, and Conn strains, have also been approved [15]. Despite these efforts, there have been IB outbreaks in Colombian farms with well-established vaccination programs [16]. Between 2003 and 2005, isolates corresponding to serotypes Conn and Mass, as well as four isolates with unique or indigenous genotypes, were identified in Colombia by RT-PCR and partial sequencing of HVR1 of the S1 subunit [17]. In 2016 and 2021, studies based on partial S1 sequences found that these viruses were close to the Mass vaccine strains and GI-11 lineage [18,19]. However, there is no information available about the whole genome sequence of IBV circulating in Colombia.

Analyses of the whole genome sequence are necessary to obtain a comprehensive characterization of IBV evolutionary dynamics. Nevertheless, the number of these sequences available in public repositories is still limited. In Latin America, only a few studies based on the complete genome sequences of IBV have been carried out in recent years, including those from Uruguay (GI-11 and GI-16) [7], Perú (GI-16) [8], and Costa Rica (GI-13) [20]. Although these lineages have widely circulated in South America, generating economic losses in the poultry industry [21], their genomic evolution is poorly understood due to the few reports in the region, particularly in Colombia. To fill this gap in knowledge, this study aims to characterize the complete genome of IBV affecting poultry producing farms in the country. The main contribution of the present study is to report, for the first time, the whole-genome sequence of field IBV isolates obtained in Colombia and their evolutionary relationships within the context of worldwide IBV genetic diversity.

## 2. Materials and Methods

### 2.1. Biological Samples and Virus Isolation

Seven IBV isolates from the Animal Virology Laboratory collection at the Universidad Nacional de Colombia were fully sequenced. These isolates were obtained in previous studies performed during 2012 and 2013 on farms with a history of respiratory disease compatible to IB. The farms were in a region with a high poultry population density (Cundinamarca, Colombia, South America). At the time of sampling, the birds had clinical signs and lesions well matched with respiratory disease (Table 1). Tissue samples and tracheal swabs were collected. Viral isolation was attempted from these samples through three blind passages in 10-day-old embryonated Specific Pathogen Free (SPF) chicken eggs. Virus infected allantoic fluid was collected and stored at −70 °C [16,22]. Later, stocks from viral isolates were prepared in embryonated SPF chicken eggs, evaluated by Haemagglutination assay (HA) and RT-PCR, and stored at −70 °C until viral RNA extraction. To measure HA titers, two-fold serial dilutions of IBV isolates were placed in V bottom 96 well plates. Fifty µL of 0.75% red blood cells suspension (RBCs) were added to every well and incubated at room temperature for 40 min. The HA titer was red when the RBCs in the RBCs control had formed a solid button in the bottom of the well.

### 2.2. RNA Extraction, cDNA Synthesis, and PCR Amplification

Viral RNA was obtained with the High Pure Viral Nucleic Acid Kit^®^ (Roche Diagnostics GmbH, Penzberg, Germany) from 200 µL of virus isolate, following the manufacturer’s instructions, in a final volume of 50 µL. cDNA synthesis was carried out from 10 µL of RNA at a concentration of 100 ng/µL, using the High Capacity cDNA Reverse Transcription Kit^®^ (Applied Biosystems, Waltham, MA, USA), with a final reaction volume of 20 µL. The reaction mixture of 10 µL contained: 50 U of RT, 20 U of RNase inhibitor, 4 mM of dNTP (100 nM), and 1.0 µL of random primers (10×). Reverse transcription was carried out with the following thermal profile: 25 °C for 10 min, 37 °C for 120 min, and 85 °C for 5 min. IBV confirmation was performed by PCR, targeting the UTR-3’ regions (UTR1–GCTCTAACTCTATACTAGCCTATACTAGCCTAT and UTR2–AAGGAAGATAGGCATGTAGCTT) [23], using the Platinum™ Taq DNA Polymerase (Thermo Fisher Scientific Inc., Waltham, MA, USA). The amplification reaction was performed under the following conditions: 94 °C initial denaturation for 5 min, 35 cycles of 94 °C for 30 s, 56 °C for 45 s, 72 °C for 2 min, and 72 °C final extension for 5 min [22].

### 2.3. Illumina Sequencing and Genome Assembly

Libraries were prepared with the Nextera XT DNA Sample preparation kit^®^ (Illumina, San Diego, CA, USA) with 1 ng of pure DNA. Quantification of libraries before sequencing was carried out by a Qubit dsDNA HS Assay Kit^®^ (Invitrogen, Waltham, MA, USA) and the starting material was diluted in 10 nM Tris-HCl, pH 7.5–8.5. A multiplexed library pool was created with 10 µL of each diluted library. Illumina MiSeq (Illumina, San Diego, CA, USA) platform was used for sequencing at the Translational Genomics Core, Louisiana State University Health Sciences Center-New Orleans.

Raw reads were evaluated with both FastQC and the quality assessment module of the CLC Genomics Workbench software version 22.0 (QIAGEN Aarhus A/S). Reads were filtered (for length, quality, and ambiguity) with Trimmomatic [24]. De novo assembly was performed with the filtered reads, using two assemblers: (i)-the de novo assembly module of CLC Genomics Workbench version 22.0 (QIAGEN Aarhus A/S) with default parameters and a minimum contig coverage of 10; and (ii)-the iterative short-read genome assembly module of SPAdes version 3.15.4 [25]. Contigs from both assemblies were analyzed and aligned for positioning and direction resolution, using NC_001451 (GCA_000862965.1) as a reference genome. The longest contigs with the highest coverage were selected for genome scaffolding and final sequence generation.

### 2.4. IBV Maximum Likelihood Phylogenetic Analyses

All complete IBV genomes and S1 gene sequences (>1000 nt in length) in GenBank (by March 2021), with available sampling date and location were downloaded and included in the analysis, resulting in a final data set of 431 complete IBV genomes, and 1972 S1 sequences from 26 and 39 countries, respectively. The sampling dates of the sequences retrieved from Genbank spanned a period of 80 years (1940 to 2020). Nucleotide sequences were aligned using MAFFT v7.467 software [26] and subsequently subjected to Maximum Likelihood (ML) phylogenetic analysis. ML phylogenetic trees were inferred with IQ-TREE 1.6.1 software [27] under the best-fit model of nucleotide substitution selected using the ModelFinder application and edited with Interactive Tree Of Life (iTOL; http://itol.embl.de) version 6.5.4 [28]. Branch support was assessed by the approximate likelihood-ratio test based on a Shimodaira–Hasegawa-like procedure (SH-aLRT) with 1000 replicates.

### 2.5. Open Reading Frames (ORFs) Prediction and Recombination Analysis

Sequences that corresponded to whole genomes (V3, V6, V8, and V10) were subjected to Open Reading Frames prediction using three tools: (i)- Find Open Reading Frames module of CLC Genomics Workbench version 22.0 (QIAGEN Aarhus A/S); (ii)- ORF Finder online platform [29]; and (iii)- BLASTn paired alignments [30], using individual sequences for ORFs 1ab, S, 3a, 3b, E, M, 4b, 4c, 5a, 5b, N, and 6b available in Genbank.

Recombination analysis was performed using Bootscan analyses as implemented in Simplot v3.5.1 [31]. S1 sequences were analyzed using a window size of 200 bp, a step size of 50 bp, and the prototype strains proposed by Valastro et al. [5] as reference sequences. The complete genomes generated in this work were also analyzed by Bootscan with a window and step sizes of 500 bp and 100 bp, respectively, and using the complete genomes available in Genbank as putative parental strains (Table 2).

To validate the recombination results obtained for V6 and V8 assembled genomes, we mapped V6 and V8 clean reads onto their corresponding assemblies with the CLC GWB resequencing tool, to assess read depth and mapping quality, before and after the putative recombination sites, in windows of 100 and 200 nucleotides. Read mapping parameters were mismatch cost 2, insertion cost 3, deletion cost 3, length fraction 0.8 and similarity fraction 0.8. In addition, we identified the proportion of reads that spanned the recombination boundaries over 50 nucleotides in both directions.

## 3. Results

### 3.1. Strains and Genome Sequencing

A total of seven IBV genomes from V2, V3, V5, V6, V8, V9, and V10 were sequenced and assembled. These isolates were obtained from trachea, tracheal swabs, and kidney from four farms in an important poultry-producing area in the country (Figure 1).

On average, 87% of reads passed the quality trimming thresholds, including only reads above 70 bp in the analysis. On average, 59.15% of the reads that generated any contigs, contributed to IBV assembly, while 12.26% were *Gallus* reads. The remaining reads contributed to the assembly of contigs that upon BLAST search corresponded to bacteria or other viral species.

Complete IBV genomes could only be assembled for four of the samples (V3, V6, V8, and V10) for which assembly sizes ranged from 27,532 nt in V6 to 27,625 nt in V3. For the remaining three samples (V2, V5, and V9), partial genomes were assembled, lacking on average the last 2500 nucleotides (Appendix A). These partial genomes included the complete S protein gene.

### 3.2. Phylogenetic Analysis of IBV

The initial classification into genotypes and lineages was based on the S1 subunit ML phylogenetic analysis, using as a reference the classification of the prototype strains proposed by Valastro et al. [5]. Isolates V2, V3, V5, V6, V9, and V10 were clustered into genotype I lineage 1 (GI-1), whereas V8 were clustered into GVI-1 (Figure 2).

Subsequently, the ML phylogenetic analysis of the S1 subunit sequences on the GI-1 lineage (Figure 3) showed that while isolates V2, V3, V5, V9, and V10 branched in a highly supported (SH-aLRT = 99%) clade that also included the vaccine strain Ma5 (KY626045), isolate V6 segregated into a clade (SH-aLRT = 94%) alongside the Mass 1941 vaccine strain (GQ504725).

To consider the evolutionary information contained throughout the IBV full-length genomes, a phylogenetic analysis was also performed, using complete IBV genome sequences (*n* = 438). As for the S1 subunit inference for the GI-1 lineage, isolates V2, V3, V5, V9, and V10 group closely related to the vaccine variant Ma5 2016 (KY626045). Meanwhile, strain V6, which in the S1 subunit phylogeny was closely related to the original Mass 1941(GQ504725) vaccine strain, branched into a cluster alongside sequences from Peafowl (AY641576) and Mass41 (FJ904721) vaccine strains. On the other hand, based on its complete genome sequence, the V8 strain grouped into a clade that also comprised sequences from Peru (MH878976), Uruguay (MF421320 and MF421319), Italy (KP780179), and China (JX195178 and JX195177), which under the classification based on S1 belong to GI-11 and GI-16 lineages (Figure 4). Hence, this result is not in agreement with the V8 classification established when only the S1 subunit sequence was analyzed. Table 3 shows the results of genotypes, lineages, and nearest sequences found when the S1 subunit and the whole genome were analyzed.

### 3.3. ORF Prediction and Recombination Analysis

The analysis of whole genome sequences based on ORF prediction results showed that IBVs from this study shared ORFs: 1ab, S, 3a, 3b, E, M, 5a, 5b, and N. However, V6 genome had three additional ORFs: 4b, 4c, and 6b, while V8 only had the addition of ORF 4b (Figure 5a).

All strains analyzed in this study show a non-recombinant profile in the S1 region. The complete genome recombination analysis suggests that V2, V3, V5, V9 and V10 strains do not present recombination events, clustering in all its extension with reference strains of the GI-1 lineage. Furthermore, this analysis indicates that V6 and V8 present mosaic genomes. The V6 genome involves at least two lineages: GI-1 and GI-16 (Figure 5b), while the V8 genome likely involves regions of four different IBV lineages: GI-1, GI-16, GI-17, and GVI-1 (Figure 5c). By analysing the Bootscan plots, we were able to narrow the recombination breakpoints to approximately the position 7084 for V6, and around positions 4798 and 20,119 for V8.

After mapping V6 clean reads to the assembled V6 genome, we identified read coverages of 529 and 533, in a window of 100 bp before and after the position 7084. There were no areas of low coverage, and mapping quality was >32 in a window of 200 nucleotides up and downstream of this site. At least 49% of mapped reads spanned the presumed recombination site over 50 nucleotides in both directions, confirming overlapping read coverage across V6 recombination boundaries. For V8 clean read coverage was 3683 before, and 3679 after the first putative recombination site (4798). Over 61.1% of the reads spanned this first breakpoint with high mapping quality and at least 50 nucleotides in both directions, serving as anchors of the recombination. The region around nucleotide 21,119 had an average read coverage of 4480 before and 4131 after that point, with 49.7% of paired reads spanning this position on both sides.

## 4. Discussion

In this study, we describe for the first time the complete genome of IBV from isolates of poultry farms in a region with a high avian population density in Colombia, evidencing the circulation of GI and GVI genotypes. It should be noted that research carried out since the first report of the virus in the country in 1963 [14] has focused on the analysis of partial sequences of the HVR-1 of S1, finding a high genetic similarity with Mass and Conn serotypes and with vaccine strain H120 [17]. Similarly, more recent studies are based only on partial S1 sequences, showing phylogenetic closeness to G1-1 Mass vaccine strains and to G1-11 lineage [18,19].

Based on complete S1 phylogenetic analyses, we found that six of the seven isolates included in the study clustered in the GI-1 lineage and corresponded to the Mass serotype. This serotype causes respiratory disease, and it is distributed worldwide due to its wide use in the design of live attenuated vaccines [5]. According to this phylogenetic inference, while V6 strain is closely related to the Mass 1941 vaccine strain, isolates V2, V3, V5, V9, and V10 are closely related to the variant vaccine strain Ma5 2016 (GI-1) obtained from live attenuated vaccines available in Latin America [32], which are also used in Colombia; therefore, this result was expected. Furthermore, in Colombia the circulation of field IBV with a high similarity to vaccine strains has been reported, particularly related to the Massachusetts H120 vaccine strain [17].

In South America most of the vaccination schemes are based on Mass strains (GI-1), even though it has been reported that the use of these vaccines does not generate cross-protection against different serotypes nor does it prevent adverse effects from field infection [33]. Due to the pathogenicity of emerging variants of IBV, epidemiological data of both genetic and antigenic characterization of viruses are essential to adjust prevention and control strategies to reduce the negative impact of IB on production [34,35,36].

Analysis of the complete S1 sequences from isolates V2 and V3 showed that these were identical, despite coming from samples of birds at different ages, belonging to different production systems (breeders and broilers) and isolated on farms located far from each other. Interestingly, even though V2 and V3 sequences are phylogenetically related to vaccine strains, they were obtained from birds with respiratory clinical signs. Therefore, this could be the result of a complex mixture of components that involve viral genotypes, immunological, and host related factors. In addition, other respiratory pathogens may generate clinical presentation resembling IB, such as lentogenic Newcastle disease virus, low pathogenicity strains of avian influenza, infectious laryngotracheitis virus, metapneumovirus, adenovirus, and *Avibacterium paragallinarum,* among others.

Another interesting finding in this study that has raised concerns in relation to its possible source, is the period between vaccination and virus isolation. It has been reported that vaccine strains can be detected 7 to 21 days post vaccination [37]. This period was consistent with what was detected in broilers, but not in breeders, in which isolation was performed seven weeks after vaccination, pointing to a possible viral persistence of more than one month after the last live attenuated vaccine administration. This is consistent with the suggestion that, vaccine strains could persist in the field and cause disease, as has been proposed by some authors [12,38].

Even though isolates V5, V6, and V10 were obtained from tracheal swabs of birds on the same farm, V5 and V10 clustered together in the S1 phylogenetic tree and are closely related to the Ma5 vaccine strain, while V6 is more closely related to the Mass41 vaccine variant. This could indicate the co-circulation in Colombia of different IBV strains in the same production unit as has been reported by previous studies [37,39].

Interestingly, based on the phylogenetic analysis of its S1 region, isolate V8 showed a different outcome and clustered in the GVI-1 lineage, which includes sequences from Asian strains reported in China [40], Republic of Korea [41], Japan [42], India [43], and Vietnam [44]. Despite being considered a geographically confined lineage in Asia, Ren et al. [45] proposed the hypothesis that this lineage originated in Colombia due to the high genetic similarity of the S1 complete sequences from Asian strains (classified in GVI-1) and Colombian S1 partial sequences obtained previously [17]. Our complete S1 sequence analysis results provide additional evidence that further support this hypothesis.

Complete genomes allow a deeper understanding of the phylogenetic evolution of variants and the recombination events between strains. Phylogenetic analysis of the whole genome IBV sequences revealed that the isolate V8 group closely related to a Peruvian sequence classified in the GI-16 lineage [8], as well as to sequences obtained in Uruguay [7], Italy [46], and China [47], all belonging to GI-11 and GI-16 lineages. Previous studies determined that lineages GI-11 and GI-16 have differences in the S1 sequence, while the rest of their genomes are mostly similar [43,44], which agrees with the phylogenetic relationships obtained here based on the analysis of complete genomes [21,47]. In addition, it is notable how these lineages are phylogenetically distant from the Mass serotype, which is routinely used in vaccination protocols in Latin America. The phylogenetic analysis of the whole genome sequences also suggests that isolate V6 is closely related to a Mass-like sequence obtained from a wild turkey in China (Peafowl, AY641576) [48]. These results could indicate a recombination process between two or more strains with different origins that leads to the emergence of novel field variants [49] with chimeric genomes [7,50,51]. Our analysis confirmed that V6 and V8 assembled genomes were true recombinants since there were no areas of low mapping around the putative recombination sites and high proportion of reads spanned both directions of the recombination boundaries with high quality mapping. Further studies are needed to characterize genetic and antigenic profiles of circulating isolates in the field and to update the strains with which vaccines should be designed.

## 5. Conclusions

In conclusion, in this study we report for the first time the whole genome sequence of IBVs in Colombia, contributing to the knowledge of coronaviruses in Latin America and the rest of the world. It is necessary to continue updating the genomic information due to the constant evolution of IBV, since the results obtained in this study correspond to samples obtained between 2012 and 2013. Recombination analysis for IBV have traditionally focused on S1 sequences. Although none of the complete S1 sequences generated in this study presented a recombinant profile, we observed recombination events in other regions of the genome. Recombination analysis requires complete S1 sequences and the lack of this type of information for some lineages reinforces the need to know the sequences of complete IBV genomes that are circulating worldwide. Whole genome sequences make it possible to determine the large number of events that led to the appearance of IBV variants. However, there is a great need for further studies to establish the relevance of recombination events in viral pathogenesis and/or immune response evasion and how this might affect the protection that current vaccines are giving to vaccinated animals. This information is essential to improve control strategies to minimize the health and economic impact related to IBV infections on the field.

## Figures and Tables

**Figure 1 viruses-14-02562-f001:**
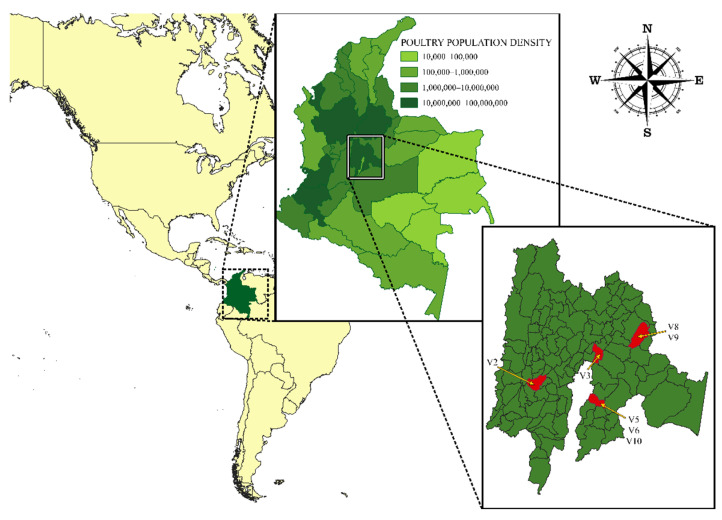
Geographical location of the farms from which the Infectious Bronchitis Virus (IBV) isolates included in the study were obtained. The lower right image shows the exact location of the farms and the seven IBV isolates (V2, V3, V5, V6, V8, V9, and V10).

**Figure 2 viruses-14-02562-f002:**
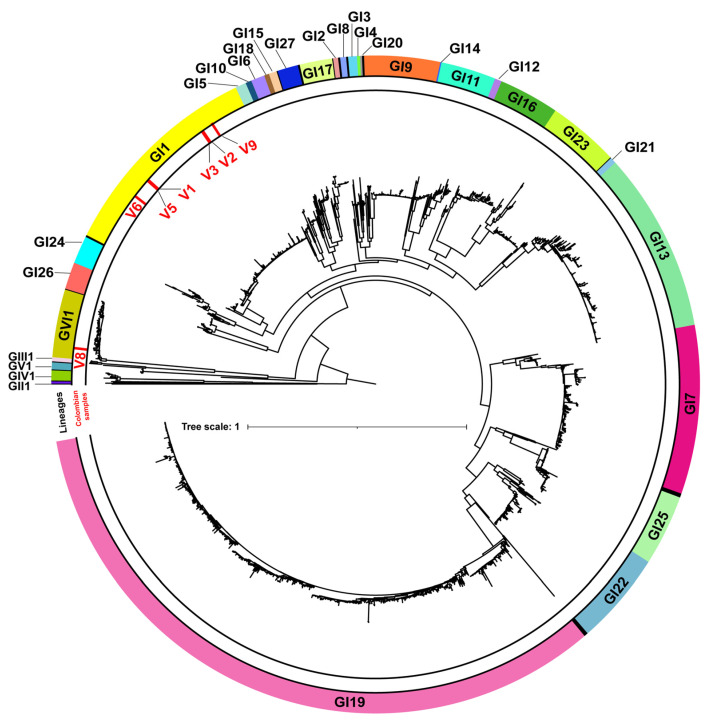
Maximum-likelihood (ML) phylogenetic tree of IBV S1 Colombian sequences (*n* = 7), plus S1 IBV worldwide gene sequences (*n* = 1972). The circular band of colors around the tree indicates the genotype and lineage of each clade. Black bands indicate unique variants as designated by Valastro et al. 2016. Red lines in the second layer (outer to inner) indicate the placement of the Colombian samples. The SH-aLRT support values were >99% for all the genotypes and defined lineages. The tree was midpoint rooted and the length of the branches are drawn to scale with the bar at the middle indicating nucleotide substitution per site.

**Figure 3 viruses-14-02562-f003:**
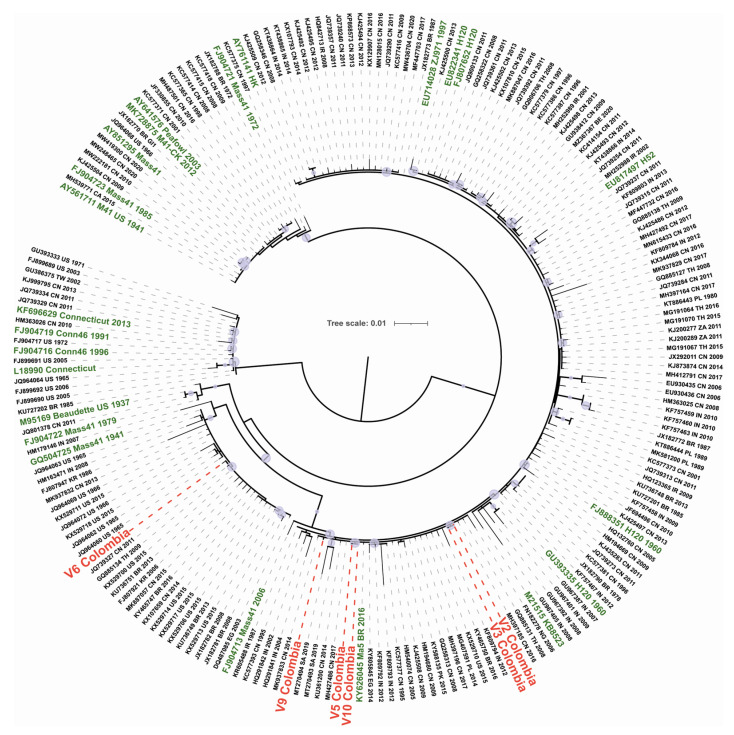
Maximum-likelihood phylogenetic tree of Infectious Bronchitis Virus (IBV) S1 gene sequences from the GI-1 clade (*n* = 203). Red labels correspond to IBV Colombian sequences and green labels to vaccine strains. Circles at the nodes denote SH-aLRT branch support values larger than 80%. The tree was midpoint rooted and the length of the branches are drawn to scale with the bar at the middle indicating nucleotide substitution per site.

**Figure 4 viruses-14-02562-f004:**
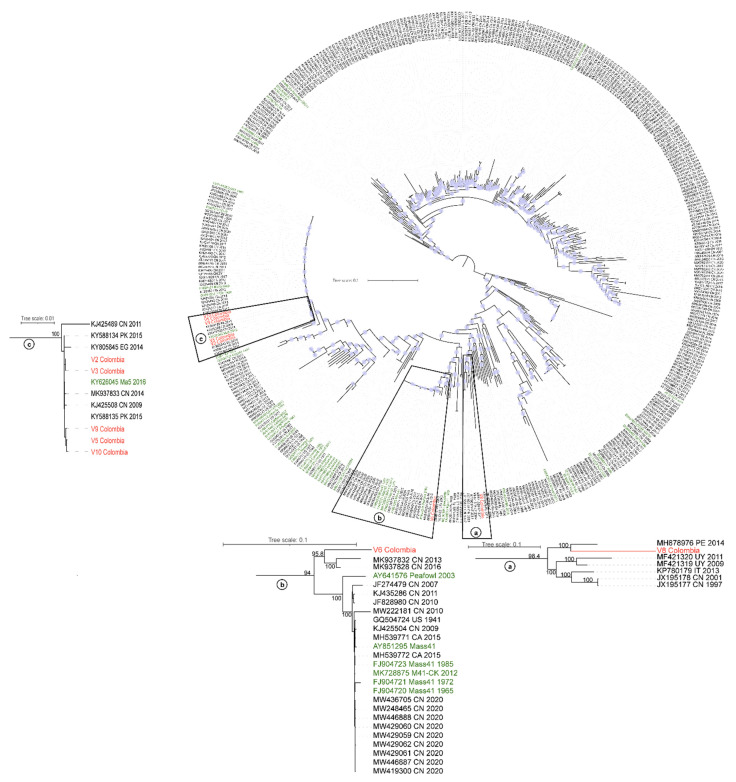
Maximum-likelihood phylogenetic tree of IBV complete genomes (*n* = 438). Red labels correspond to IBV Colombian sequences and green labels to vaccine strains. Circles at the nodes denote SH-aLRT branch support values larger than 80%. The tree was midpoint rooted and the lengths of the branches are drawn to scale with the bar at the middle indicating nucleotide substitution per site. Clusters that contain a) V8 b) V6 and c) V2, V3, V5, V9 and V10 Colombian strains are highlighted and magnified. SH-aLRT branch support values are indicated at key nodes and the branch lengths are drawn to scale with the bar at the top indicating nucleotide substitution per site.

**Figure 5 viruses-14-02562-f005:**
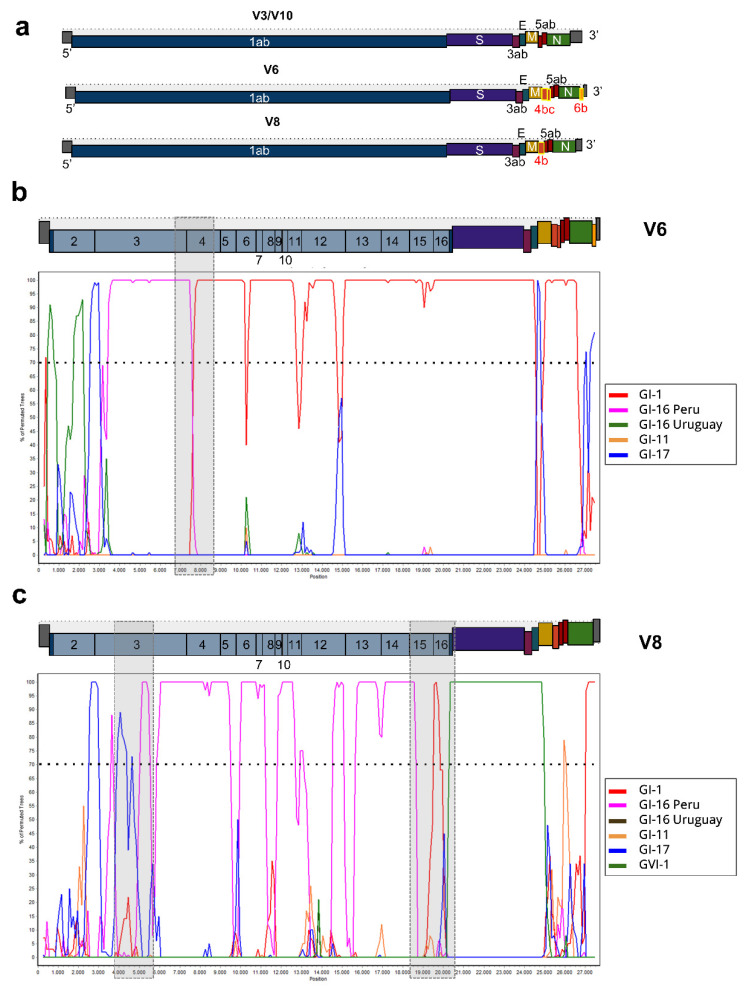
ORF prediction and recombination analyses. (**a**) Genomic organization of V3, V10, V6, and V8. Red labels and yellow boxes highlight variation regions in detected ORFs. (**b**) and (**c**) Bootscan analysis (Simplot 3.5.1) of V6 and V8 whole genome sequences, respectively (windows size: 500 bp, step size: 100 bp). The dotted lines indicate a bootstrap value of 70%, and gray boxes indicate regions with recombination events.

**Table 1 viruses-14-02562-t001:** Clinical history of poultry associated with Infectious Bronchitis Virus (IBV) isolates obtained between 2012 and 2013.

Virus	HA Titer	Tissue	Productive Unit	Vaccine Strain	Age at Last Vaccination	Time between Isolation and Last Vaccination	Clinical Signs
V2	1/2048	Trachea	Breeder	Ma5/H120	26 w	7 w	Respiratory signs and trachea lesions.
V3	1/2048	Trachea	Broiler	M48/Ma5	16 d	7 d	Ocular discharge, respiratory signs, and trachea and lung lesions.
V5 V6 V10	1/4096 1/4096 1/4096	Tracheal swab	Broiler	Ma5	5 d	15 d	Nasal-ocular discharge, swollen head, and trachea lesions.
V8	1/256	Trachea	Broiler	Ma5	14 d	15 d	Ocular discharge and kidney lesions.
V9	1/4096	Kidney					

**Table 2 viruses-14-02562-t002:** Reference sequences available in Genbank used as putative parental strains.

GenBank Accession Number	Strain Name	Country of Origin	Genotype and Lineage
MH878976	VFAR-047	Peru	GI-16
MF421319	UY/09/CA/01	Uruguay	GI-16
MF421320	UY/11/CA/18	Uruguay	GI-11
MK957245	PR01	Brazil	GI-11
MK953937	Brazil/SP55	Brazil	GI-11
MK957244	PR05	Brazil	GI-11
KX258195	23/2013	Brazil	GI-11
MN757859	CK/CR/1160/16	Costa Rica	GI-17
AY85129	Mass 41	USA	GI-1
FJ807652	H120	USA	GI-1
MK574042	ck/CH/LHB/110615	China	GVI-1

**Table 3 viruses-14-02562-t003:** Genotypes, lineages, and nearest sequences from IBV isolates obtained in this study.

Virus	S1 Subunit					Whole Genome				
	Nearest Sequence	Genotype-Lineage	Percentage of Identity	Query Cover	Country	Nearest Sequence	Genotype-Lineage	Percentage of Identity	Query Cover	Country
V2	MH427486	GI-1	99.94	100%	China	KY626045	GI-1	99.98	100%	Brazil
V3	MH427486	GI-1	99.94	100%	China	KY626045	GI-1	99.97	99%	Brazil
V5	MH427486	GI-1	99.94	100%	China	KY626045	GI-1	99.98	100%	Brazil
V6	MK937828	GI-1	99.88	100%	China	OM912698	GI-1	96.96	99%	Mexico
V8	KC692307	GVI-1	94.51	100%	China	MH878976	GI-11	91.53	99%	Peru
V9	MH427486	GI-1	99.94	100%	China	KY626045	GI-1	99.97	100%	Brazil
V10	MH427486	GI-1	99.88	100%	China	KY626045	GI-1	99.99	99%	Brazil

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
