# Peer review of "New Insights into Avian Infectious Bronchitis Virus in Colombia from Whole-Genome Analysis"

_viruses, 2022, doi:10.3390/v14112562_

Round 1

Reviewer 1 Report

In this manuscript, the authors sequence and analyze the full genomes of 7 IBV isolates from Colombia South America. The isolates originated from 14 different poultry farms with a history of respiratory disease. And although the viruses were isolated approximately 10 years ago (an eternity for IBV evolution), the full-length genomes do add to our understanding of the evolutionary relationships among IBV types in S. America and worldwide.

The manuscript is well written and the methods are appropriate but the authors need to describe how they determined that only one IBV type was in the sample sequenced by NGS. This is particularly important since the Illumina NGS platform was used to sequence the genomes. Illumina has low error rates but typically yields short reads that then must be assembled using bioinformatics. If more than one IBV type is in the sample, it is possible to have assemblies that contain sequences from both types giving the appearance of recombination within the viral genome. Overall, the manuscript has clearly defined objectives and the conclusions are not overstated and justified by the data.

The following suggestions also need to be considered:

-Table 1, HA title should be HA titer.

-Figures 2 and 3 show analysis of S1 nucleotide sequences. Suggest analyzing the AA sequence of S1 for the isolates to verify their groupings. This is particularly important for isolate V8 which grouped in the GVI-1 lineage using S1 sequence but appeared to be in the GI-11 lineage when the full genome was analyzed.

-Samples analyzed in this study were taken for virus isolation at 7 and 15 days post-vaccination for all but 1 sample (V2- 7 weeks). It is not unusual for vaccine strains to still be replicating in birds for up to 2 weeks and they will grow preferentially in eggs making them the most likely virus to be isolated. Indeed 6 of the 7 viruses sequenced were GI-1 (Mass) which appear to be vaccine type viruses. Although the viruses were likely vaccine origin virus, they were not put back into birds to determine their ability to cause disease, which would have verified this suspected result. That said, MA5-like viruses are unique may well be vaccine origin. It would be good to point this out in the discussion.

-Line 253; The sentence that says ‘reporting phylogenetic closeness to Mass vaccine strains, GI-1 and GI-11 lineages’ is confusing since the GI-11 lineage are not Mass type viruses. Suggest rewording.

-Line 268; the Sentence ‘In contrast, the use of homologous vaccines has been reported to lead to the emergence of variants that evade specific immunity” should be deleted or explained further. Although the authors reference the statement [30], it isn’t that simple. There are many factors (including homologous and heterologous vaccination) that contribute to the evolution of IBV and RNA viruses in general. It would take several pages to describe the phenomenon in total thus, it is recommended to delete the sentence for that reason and because looking at the evolution of IBV isolated from vaccinated birds was not the objective of this study.

-Line 278, indicate that respiratory pathogens other than IBV could also be involved.

Author Response

In this manuscript, the authors sequence and analyze the full genomes of 7 IBV isolates from Colombia South America. The isolates originated from 14 different poultry farms with a history of respiratory disease. And although the viruses were isolated approximately 10 years ago (an eternity for IBV evolution), the full-length genomes do add to our understanding of the evolutionary relationships among IBV types in S. America and worldwide.

Answer: We appreciate the comments and acknowledge the limitations of the study, such as the period analyzed. However, we consider that despite the above, the study remains relevant given the lack of information related to the whole genome sequences and S1 complete nucleotide sequences in Colombia. It is important to point out that the isolates from this study were focused on knowing the characteristics of the coronaviruses that caused infections and clinical signs in poultry despite vaccination programs implemented in the country. We believe that it is important to provide information on the characteristics of the circulating viruses for further studies with more current surveillance samples.

The manuscript is well written and the methods are appropriate but the authors need to describe how they determined that only one IBV type was in the sample sequenced by NGS. This is particularly important since the Illumina NGS platform was used to sequence the genomes. Illumina has low error rates but typically yields short reads that then must be assembled using bioinformatics. If more than one IBV type is in the sample, it is possible to have assemblies that contain sequences from both types giving the appearance of recombination within the viral genome. Overall, the manuscript has clearly defined objectives and the conclusions are not overstated and justified by the data.

Answer: An additional paragraph was included in Materials and methods to describe how we determined that only one IBV type was in the sample sequenced by NGS (L169-179). Two paragraphs were also included in results clarifying these doubts (L264-289).

The following suggestions also need to be considered:

-Table 1, HA title should be HA titer.

Answer: This change was done.

-Figures 2 and 3 show analysis of S1 nucleotide sequences. Suggest analyzing the AA sequence of S1 for the isolates to verify their groupings. This is particularly important for isolate V8 which grouped in the GVI-1 lineage using S1 sequence but appeared to be in the GI-11 lineage when the full genome was analyzed.

Answer: We appreciate the reviewer's suggestion and understand that phylogenetic inferences can be made with both nucleotide and amino acid sequences. However, the optimal choice depends on the level of evolutionary relationship being investigated. The phylogenetic inference of closely related Operational Taxonomic Units (OTUs), as is our case, is commonly done based on nucleotide sequences because it allows the detection of synonymous changes. The nucleotide sequences of a pair of homologous genes have higher information content than the amino acid sequences of the corresponding proteins leading to a better resolution of the tree. Therefore, we chose to keep our phylogenetic analysis as it was done, based on nucleotide sequences, since that way we obtain the highest phylogenetic resolution. If deeper evolutionary relationships are being studied (which is not our case), then analysis of amino acid sequences is more appropriate because changes occur more slowly thus avoiding the saturation of transitions and/or third-codon positions. On the other hand, genetic typing based on the S1 gene is not expected to be consistent with clusters based on full-length sequences. The complexity and number of variable or informative sites in the multiple sequence alignment, as well as recombination, are important factors that impact the clustering and therefore can generate different tree topologies.

[1]T. A. Brown, “Molecular Phylogenetics,” 2002.

[2]N. H. P. Nicholas H. Barton, Derek Eg Briggs, Jonathan A. Eisen, David B. Goldstein, “Phylogenetic Reconstruction,” in Evolution (Book), Cold Spring Harbor Laboratory Series, 2008, p. 55.

[3]A. E. Gorbalenya and C. Lauber, “Phylogeny of Viruses,” in Reference Module in Biomedical Sciences, January, Elsevier, 2017.

-Samples analyzed in this study were taken for virus isolation at 7 and 15 days post-vaccination for all but 1 sample (V2- 7 weeks). It is not unusual for vaccine strains to still be replicating in birds for up to 2 weeks and they will grow preferentially in eggs making them the most likely virus to be isolated. Indeed 6 of the 7 viruses sequenced were GI-1 (Mass) which appear to be vaccine type viruses. Although the viruses were likely vaccine origin virus, they were not put back into birds to determine their ability to cause disease, which would have verified this suspected result. That said, MA5-like viruses are unique may well be vaccine origin. It would be good to point this out in the discussion.

Answer: We understand the reviewer's concern and appreciate the suggestion; however, the presentation of clinical signs despite having a vaccination schedule is striking. This is a problem that is seen on farms frequently, so the aim of this study was to describe the genetic characteristics of the isolated viruses, considering the possibility of identifying variants of the virus, responsible for the clinical signs and lesions. We believe that it is important to provide information on the characteristics of the circulating viruses in these populations that present respiratory symptoms despite being vaccinated. The possible co-circulation in Colombia of different IBV strains in the same production unit agrees with results from previous studies. This fact is showed in the discussion section. This study was not aimed at a characterization of pathogenicity; however, the viral isolates obtained will allow the selection of candidates for antigenic and pathogenicity analysis in future studies.

-Line 253; The sentence that says ‘reporting phylogenetic closeness to Mass vaccine strains, GI-1 and GI-11 lineages’ is confusing since the GI-11 lineage are not Mass type viruses. Suggest rewording.

Answer: This sentence was rewrote between Lines 328-330.  

-Line 268; the Sentence ‘In contrast, the use of homologous vaccines has been reported to lead to the emergence of variants that evade specific immunity” should be deleted or explained further. Although the authors reference the statement [30], it isn’t that simple. There are many factors (including homologous and heterologous vaccination) that contribute to the evolution of IBV and RNA viruses in general. It would take several pages to describe the phenomenon in total thus, it is recommended to delete the sentence for that reason and because looking at the evolution of IBV isolated from vaccinated birds was not the objective of this study.

Answer: We appreciate this recommendation and the sentence was deleted.

-Line 278, indicate that respiratory pathogens other than IBV could also be involved.

Answer: Respiratory pathogens were included in L355-358.

Reviewer 2 Report

The submission by Ramirez-Nieto, et al, New insights into Avian Infectious Bronchitis Virus in Columbia from whole -genome analysis, is a relatively well written article that describes whole genome sequencing (WGS) and subsequent analysis of seven avian infectious bronchitis virus (IBV) isolates from samples obtained from poultry farms located in central Columbia.  Once the overall story of the paper is clarified and conclusions are drawn from the data presented the submission is recommended for publication with revisions.

General comments:

It is obvious this submission was directly translated into English and not edited for either sentence structure or English language conventions.  There are occasions where the Spanish has not been removed (y is used instead of and ). Several sentences are awkward for the reader and need clarification.  The author tends to over-use parentheses and should check each instance for correctness.

The seven samples used for this analysis are from a previous study and were collected in 2012 and 2013.  The inclusion of more current surveillance samples and a larger data set will strengthen this study.

The author makes excellent use of the harmonized classification system described by Valastro, et al, 2016.

The author describes the function of the S region of the genome but does not describe the remainder of the structural and accessory proteins.  Accessory protein 6b is attributed to one sequenced isolate in the results section but is not described in the introduction as even occurring in the prototypical IBV genome. 

Specific Comments:

Line 18-20: The sentence that begins with “IB virus (IBV)…” is awkward - revise

Line 62: Consider listing the genotype G1 after “one main genotype” and the lineage numbers in like

manner – then list the names in the parentheses (GI-11 and GI-16)

Line 75-77: Sentence beginning with “Likewise, recent studies…” is awkward – revise

Line 79-81: Run on sentence

Line 83-84: Explain why you are listing the genotypes and lineages – Why is this important?

Line 91-101: Seven isolates from 14 farms?  Maybe leave out the number of farms to reduce confusion. The tissues you describe being collected does not match those described in Table 1.  What substrate were the viruses isolated in (embryonated chicken eggs? CEF cells?) You must be specific.   Which viruses were used to make stocks and why was this done?

Table 1: This table needs a major revision and clarification – only include the information that is pertinent – make it neat and easy to read.  Id: what does this mean? Titer not title.  There is no V10 although it is discussed in the remainder of the manuscript – this is a major error.

Line 110: Use decimal points (.) instead of commas (,)

Line 133: Please list the strain number associated with the GenBank accession number – GenBank needs referencing

Line 137 – 141: GenBank needs referencing: Clarify which genomes were spanning 80 years; the ones from GenBank or the ones you sequenced

Line 149 -162: Be careful of run-on sentences; consider using a table to list the sequences used as reference that would include GenBank Accession number, strain name, country of origin, genotype and lineage

Line 165 – 181: The isolated viruses discussed in the results section do not match with Table 1.

Figure 1: Lovely figure – however consider your color choices; black lines on a dark green background are tough to read

Table 2: This should be a supplemental table if used at all.

Figure 2: A lovely figure albeit your data is difficult to locate.  Try and find a way to highlight your data.

Figure 3: Another lovely figure where your data is hidden.

Table 3: Revise for clarity

Figure 4: Yet another lovely figure that hides your data – consider enlarging the pertinent sections for clarity; giving you figure 4a – the whole picture, 4b and 4c the enlarged sections

Line 231 -234: Reword this section for clarity. 

Figure 5: 5a: highlight the variations, 5b and 5c: label / highlight the regions of the Simplot analysis that are important and are discussed in the body of the manuscript – consider a way to make your legend larger

Conclusion: The conclusions you draw are not based on your data.  What does your data point to – then end with a paragraph similar to the one you have.

Author Response

General comments:

It is obvious this submission was directly translated into English and not edited for either sentence structure or English language conventions.  There are occasions where the Spanish has not been removed (y is used instead of and ). Several sentences are awkward for the reader and need clarification.  The author tends to over-use parentheses and should check each instance for correctness.

Answer: We appreciate the comments and we apologize for the mistakes. These aspects were reviewed and corrected in full document.

The seven samples used for this analysis are from a previous study and were collected in 2012 and 2013.  The inclusion of more current surveillance samples and a larger data set will strengthen this study.

Answer: We appreciate the comments and acknowledge the limitations of the study, such as the period analyzed. However, we consider that despite the above, the study remains relevant given the lack of information related to the whole genome sequences and S1 complete nucleotide sequences in Colombia. It is important to point out that the isolates from this study were focused on knowing the characteristics of the coronaviruses that caused infections and clinical signs in poultry despite vaccination programs implemented in the country. We believe that it is important to provide information on the characteristics of the circulating viruses for further studies with more current surveillance samples.

The author makes excellent use of the harmonized classification system described by Valastro, et al, 2016.

Answer: We appreciate the comments.

The author describes the function of the S region of the genome but does not describe the remainder of the structural and accessory proteins.  Accessory protein 6b is attributed to one sequenced isolate in the results section but is not described in the introduction as even occurring in the prototypical IBV genome.

Answer: Information about other structure and accessory proteins was included (L51-57).  

Specific Comments:

Line 18-20: The sentence that begins with “IB virus (IBV)…” is awkward – revise

Answer: The sentences were reviewed and corrected.

Line 62: Consider listing the genotype G1 after “one main genotype” and the lineage numbers in like manner – then list the names in the parentheses (GI-11 and GI-16)

Answer: This suggestion was done.

Line 75-77: Sentence beginning with “Likewise, recent studies…” is awkward – revise

Answer: This sentence was reviewed and rewrote for clarification.

Line 79-81: Run on sentence

Answer: These sentences were reviewed and rewrote for clarification.

Line 83-84: Explain why you are listing the genotypes and lineages – Why is this important?

Answer: The explanation was included in L96-98.

Line 91-101: Seven isolates from 14 farms?  Maybe leave out the number of farms to reduce confusion. The tissues you describe being collected does not match those described in Table 1.  What substrate were the viruses isolated in (embryonated chicken eggs? CEF cells?) You must be specific.   Which viruses were used to make stocks and why was this done?

Answer: We leaved out the number of farms (Line 108). The materials and methods section describe the tissues from which isolation was attempted, but the table lists the organs from which it was achieved. The parentheses were removed, following the suggestion to review over-use parentheses. Substrate used for viral isolation and stocks was mentioned in L111-117.

Table 1: This table needs a major revision and clarification – only include the information that is pertinent – make it neat and easy to read.  Id: what does this mean? Titer not title.  There is no V10 although it is discussed in the remainder of the manuscript – this is a major error.

Answer: We appreciate the reviewer's comments and we were also concerned about the possible error of omitting V10. However, after carefully reviewing the original document, we noted that V10 was listed together with V5 and V6 in the original Table 1. To facilitate reading the information in table 1 it was simplified.

Line 110: Use decimal points (.) instead of commas (,)

Answer: This change was done.

Line 133: Please list the strain number associated with the GenBank accession number – GenBank needs referencing

Answer: The GenBank accession number was included (L155).

Line 137 – 141: GenBank needs referencing: Clarify which genomes were spanning 80 years; the ones from GenBank or the ones you sequenced

Answer: This sentence was clarified in L162.

Line 149 -162: Be careful of run-on sentences; consider using a table to list the sequences used as reference that would include GenBank Accession number, strain name, country of origin, genotype and lineage

Answer: A table containing this information was included in the main manuscript (Table 2).

Line 165 – 181: The isolated viruses discussed in the results section do not match with Table 1.

Answer: The information was reviewed and adjusted accordingly (L199-202).

Figure 1: Lovely figure – however consider your color choices; black lines on a dark green background are tough to read

Answer: The figure was edited.

Table 2: This should be a supplemental table if used at all.

Answer: The table 2 was left as a supplemental table (Table S1).

Figure 2: A lovely figure albeit your data is difficult to locate.  Try and find a way to highlight your data.

Answer: The figure was edited.

Figure 3: Another lovely figure where your data is hidden.

Answer: The figure was edited.

Table 3: Revise for clarity

Answer: Table 3 was revised for clarity.

Figure 4: Yet another lovely figure that hides your data – consider enlarging the pertinent sections for clarity; giving you figure 4a – the whole picture, 4b and 4c the enlarged sections

Answer: The figure was edited.

Line 231 -234: Reword this section for clarity.

Answer: this section was reworded (L300-304). 

Figure 5: 5a: highlight the variations, 5b and 5c: label / highlight the regions of the Simplot analysis that are important and are discussed in the body of the manuscript – consider a way to make your legend larger

Answer: The figure was edited.

Conclusion: The conclusions you draw are not based on your data.  What does your data point to – then end with a paragraph similar to the one you have.

Answer: Conclusions were reviewed.

Reviewer 3 Report

The authors determined complete genome sequences of 7 IBV isolates obtained during 2012-2013 in Colombia and performed phylogenetic and in silico recombination analyses. Five isolates (V2, V3, V5, V9 and V10) were related to vaccine strains without recombination events but two (V6 and V8) showed mosaic genomes reflecting multiple recombination events. The authors verified persistence infection of vaccine strains and increased virulence of vaccine strains and presence of different IBV in the same flock.

General comments

The genomic information of IBV isolates in Colombia is valuable, but most of results are similar to previous reports of others.

Specific comments

1. I can’t find accession numbers of the genome sequences determined in this study. They are required for further evaluation of manuscript and suggestion.

2. Please briefly describe how to measure the HA titers of IBV isolates in M & M.

3. Table 2 (final assembly size)

 - (pb)? Base pair? Nucleotide?

 - Unusual size of IBV genome: V2, V5, and V9. Please check errors in NGS by checking possible lost regions with PCR and Sanger method sequencing. Only after confirmation the authors can use ‘complete genome’ in the manuscript.

4. Table 3: Why don’t you compare 1ab sequences with others in the GenBank database? Although recombination events occur genome-wide the nonstructural proteins in 1ab tend to evolve together. Thus, trial to find most identical 1ab is important and, if you find something luckily, it is better to include their genome sequences in recombination analysis [Arch Virol (2017) 162:1237–1250, Comparative  genomics  of  QX-like  infectious  bronchitis  viruses in  Korea].

5. Line 273: Are S1 sequences of V2 and V3 identical…? They showed 99.44 and 99.94 percent identities to MH427486 (Table 3).

6. Typos?

 - Table 1: HA ‘title’ to ‘titer’?

- Line 233: please delete ‘y’

Author Response

General comments

The genomic information of IBV isolates in Colombia is valuable, but most of results are similar to previous reports of others.

Answer: We appreciate the comments; however, it is worth to consider that the information available for IBV in Colombia is only based on partial S1 sequences, but this study reports for the first time the whole-genome sequence of IBVs in the country. In addition, it is important to point out that the isolates from this study were focused on knowing the characteristics of the coronaviruses that caused infections and clinical signs in poultry despite vaccination programs implemented. We believe that it is important to provide information on the characteristics of the circulating viruses for further studies with more current surveillance samples.

Specific comments

I can’t find accession numbers of the genome sequences determined in this study. They are required for further evaluation of manuscript and suggestion.

Answer: Sequences were submitted to GenBank (submission ID: 2642739).

Please briefly describe how to measure the HA titers of IBV isolates in M & M.

Answer: HA titers measure method was briefly described (L117-L121).

  1. Table 2 (final assembly size)

 - (pb)? Base pair? Nucleotide?

Answer: The final assembly size was reviewed and corrected.

 - Unusual size of IBV genome: V2, V5, and V9. Please check errors in NGS by checking possible lost regions with PCR and Sanger method sequencing. Only after confirmation the authors can use ‘complete genome’ in the manuscript.

Answer: The following paragraph was included in manuscript (L216-220): Complete IBV genomes could only be assembled for 4 of the samples (V3, V6, V8, and V10) for which assembly sizes ranged from 27,532 nt in V6 to 27,625 nt in V3. For the remaining three samples (V2, V5, and V9) partial genomes were assembled, lacking in average the last 2,500 nucleotides (Table 2). These partial genomes included the complete S protein gene.

Table 3: Why don’t you compare 1ab sequences with others in the GenBank database? Although recombination events occur genome-wide the nonstructural proteins in 1ab tend to evolve together. Thus, trial to find most identical 1ab is important and, if you find something luckily, it is better to include their genome sequences in recombination analysis [Arch Virol (2017) 162:1237–1250, Comparative  genomics  of  QX-like  infectious  bronchitis  viruses in  Korea].

Answer: We appreciate the suggestion, and therefore we decided to compare the 1ab sequences obtained in this study with sequences with significant alignment according to BLAST. For V2, V3, V5, V9, and V10, the closest sequence was KY626045.1 (strain Ma5), while for V6 it was a Mexican sequence OM912698.1 (previously classified as GI-1), and for V8 was the Peruvian sequence MH878976.1. The only sequence obtained through BLAST analysis that had not been included previously in the recombination analysis was the isolate obtained in Mexico OM912698.1 and published in April 2022, so we decided to include this in a new run for the Bootscan recombination analysis. The results showed no difference in the recombination pattern when including OM912698.1 as part of the GI-1 group or when using as a parental sequence. When selecting this sequence as a query, we found that it presented a similar pattern to sequences close to M41 (with which V6 clusters in the phylogenetic analyses). For these reasons, we decided to keep the analyses previously presented in the manuscript.

Line 273: Are S1 sequences of V2 and V3 identical…? They showed 99.44 and 99.94 percent identities to MH427486 (Table 3).

Answer: Percent identities were reviewed and corrected in Table 3.

  1. Typos?

 - Table 1: HA ‘title’ to ‘titer’?

Answer: This change was done.

- Line 233: please delete ‘y’

Answer: This change was done.

Round 2

Reviewer 2 Report

Nicely edited.  I recommend this manuscript for publication.

Reviewer 3 Report

All questions were answered.